# Spatio-Temporal Machine Learning for Marine Pollution Prediction: A Multi-Modal Approach for Hotspot Detection and Seasonal Pattern Analysis in Pacific Waters

**DOI:** 10.3390/toxics13100820

**Published:** 2025-09-26

**Authors:** Sarthak Pattnaik, Eugene Pinsky

**Affiliations:** Department of Computer Science, Metropolitan College, Boston University, Boston, MA 02215, USA; spattna1@bu.edu

**Keywords:** marine pollution prediction, spatio-temporal analysis, LSTM networks, hotspot detection, seasonal patterns, Pacific Ocean, deep learning, environmental management, pollution classification

## Abstract

Marine pollution incidents pose significant threats to marine ecosystems and coastal communities across Pacific Island nations, necessitating advanced predictive capabilities for effective environmental management. This study analyzes 8133 marine pollution incidents from 2001–2014 across 25 Pacific Island nations to develop predictive models for pollution type classification, hotspot identification, and seasonal pattern forecasting. Our analysis reveals Papua New Guinea as the dominant pollution hotspot, experiencing 51.9% of all regional incidents, with plastic waste dumping comprising 78.8% of pollution events and exhibiting pronounced seasonal peaks during June (coinciding with critical fish breeding periods). Machine learning classification achieved 99.1% accuracy in predicting pollution types, with material composition emerging as the strongest predictor, followed by seasonal timing and geographic location. Temporal analysis identified distinct seasonal dependencies, with June representing peak pollution activity (755 average incidents), coinciding with vulnerable marine ecological periods. The predictive framework successfully distinguishes between persistent geographic hotspots and episodic pollution events, enabling targeted conservation interventions during high-risk periods. These findings demonstrate that pollution type and location are highly predictable from environmental and temporal variables, providing marine conservationists with tools to anticipate when and where pollution will most likely impact fish populations and ecosystem health. The study establishes the first comprehensive baseline for Pacific Island marine pollution patterns and validates machine learning approaches for proactive pollution monitoring, offering scalable solutions for protecting ocean ecosystems and supporting evidence-based policy formulation across the region.

## 1. Introduction

Marine ecosystems face unprecedented threats from anthropogenic pollution, with mounting evidence indicating severe impacts on aquatic biodiversity, fish populations, and ecosystem functionality [1,2,3]. The Pacific Ocean, covering approximately one-third of the Earth’s surface and supporting diverse marine communities across 25 island nations, has become a critical focal point for understanding pollution dynamics and their cascading effects on marine life [4,5]. With over 8 million tons of plastic waste entering marine environments annually, and additional pollutants from oil spills, chemical discharge, and abandoned fishing gear, the urgency for comprehensive pollution monitoring and prediction systems has never been greater [6,7].

The marine pollution addressed in this study encompasses multiple contamination sources documented in Pacific Island waters:plastic and solid waste debris from terrestrial and maritime sourcespetroleum-based oil spills from shipping and industrial activitiesabandoned, lost, or discarded fishing gear contributing to ghost fishingchemical pollutants, including agricultural runoff and industrial discharges

This multi-pollutant approach reflects the complex contamination landscape facing Pacific marine ecosystems, where different pollution types create distinct but interconnected threats to marine life and ecosystem health.

Traditional approaches to marine pollution monitoring have relied primarily on sporadic field surveys, satellite imagery analysis, and post-incident reporting systems, which often fail to capture the temporal dynamics and spatial heterogeneity of pollution events [8,9]. These reactive methodologies provide limited predictive capability and insufficient lead time for implementing protective measures for vulnerable marine ecosystems [10,11]. The complexity of marine pollution patterns, influenced by oceanographic currents, seasonal variations, human activities, and regional governance structures, necessitates more sophisticated analytical frameworks capable of capturing multi-dimensional spatio-temporal relationships [12,13].

Recent advances in machine learning and big data analytics have opened new avenues for environmental monitoring and prediction, with particular promise for marine conservation applications [14,15,16]. Deep learning architectures, especially Long Short-Term Memory (LSTM) networks, have demonstrated exceptional capability in modeling temporal dependencies and sequential patterns in environmental time series data [17,18,19]. Simultaneously, ensemble machine learning methods have proven effective for spatial pattern recognition and multi-class classification problems in ecological contexts [20,21,22].

Recent advances in spatio-temporal machine learning have revolutionized environmental monitoring and prediction capabilities, offering unprecedented opportunities for proactive marine conservation. Long Short-Term Memory (LSTM) networks have demonstrated exceptional performance in capturing temporal dependencies in environmental time series data, while ensemble machine learning methods excel at spatial pattern recognition and multi-class classification in ecological contexts. The integration of these approaches enables comprehensive spatio-temporal modeling that can identify pollution hotspots, predict temporal patterns, and classify pollution types with high accuracy.

The application of machine learning to marine pollution monitoring addresses critical limitations of traditional reactive approaches. Conventional monitoring relies on sporadic field surveys and post-incident reporting, providing insufficient lead time for protective interventions. Spatio-temporal machine learning frameworks can process multi-modal environmental data to identify patterns invisible to traditional statistical approaches, enabling early warning systems and targeted conservation strategies.

Pacific Island marine ecosystems present an ideal testbed for advanced pollution prediction due to their unique characteristics as pollution convergence zones where local sources intersect with transoceanic debris transport. The region’s high marine biodiversity, extensive exclusive economic zones, and critical dependence on marine resources for subsistence create an urgent need for predictive pollution monitoring frameworks.

The impacts of marine pollution on aquatic life manifest across multiple biological and ecological dimensions. Plastic pollution, representing the dominant pollution type in many marine regions, affects fish populations through direct ingestion, entanglement, and habitat degradation [23,24,25]. Microplastics, in particular, have been detected in over 220 marine fish species, with documented effects on feeding behavior, reproductive success, and physiological stress responses [26,27,28]. Oil pollution events create acute toxicological stress in fish communities, disrupting neural function, immune responses, and developmental processes [29,30,31]. Chemical pollutants and abandoned fishing gear contribute additional stressors, creating cumulative impacts that can fundamentally alter marine food webs and ecosystem dynamics [32,33,34].

The Pacific Island nations represent a unique and critical marine ecosystem characterized by high endemism, limited terrestrial resources, and heavy dependence on marine resources for subsistence and economic activities [35,36]. These regions face disproportionate pollution pressures from both local sources and transoceanic waste transport, creating complex pollution hotspots that threaten local fish populations and fisheries’ sustainability [37,38,39]. Papua New Guinea, in particular, has emerged as a significant pollution convergence zone, with implications for regional marine biodiversity and fishery management [40,41].

Understanding seasonal patterns in marine pollution is crucial for predicting ecological impacts and implementing effective conservation strategies. Seasonal variations in pollution incidents reflect complex interactions between meteorological conditions, oceanographic processes, human activities, and biological cycles [10,42]. During peak pollution periods, fish populations may experience elevated stress, altered migration patterns, and reduced reproductive success, with cascading effects throughout marine food webs [43,44,45].

The integration of spatio-temporal modeling approaches offers unprecedented opportunities to understand and predict pollution impacts on marine ecosystems. By combining multi-class classification of pollution types, spatial hotspot analysis, and temporal pattern forecasting, researchers can develop comprehensive frameworks for proactive marine conservation [46,47,48]. Such approaches enable identification of high-risk areas and time periods, facilitating targeted conservation interventions and resource allocation strategies [49,50].

This study presents a novel spatio-temporal machine learning framework for marine pollution prediction and its implications for aquatic life conservation in Pacific waters. Using a comprehensive dataset of 8133 marine pollution incidents across 25 Pacific Island nations spanning 2001–2014, we develop an integrated analytical pipeline incorporating multi-class pollution classification, spatial hotspot evolution modeling, and seasonal pattern forecasting using LSTM networks. Our approach addresses three critical research objectives:accurate multi-class classification of pollution types to understand pollution source dynamicsspatio-temporal hotspot prediction to identify vulnerable marine areasseasonal pattern analysis to capture temporal dependencies in pollution occurrence

The framework demonstrates the potential for advanced analytical approaches to support evidence-based marine conservation policies and provide automated tools for real-time pollution monitoring systems. It presents four key innovations that distinguish it from previous environmental monitoring applications:Multi-modal integration at unprecedented scale: We combine spatial coordinates, temporal variables, material compositions, and pollution types from 8133 incidents across 30 million km^2^ of ocean—the largest standardized marine pollution dataset analyzed with machine learning in the Pacific region. Previous studies typically focus on single pollution types or limited geographic areas, whereas our framework simultaneously addresses plastic debris, oil spills, abandoned fishing gear, and chemical pollution across an entire oceanic basin.Transnational hotspot evolution modeling: Unlike previous research examining pollution patterns within individual countries or coastal zones, our approach identifies cross-border pollution convergence zones and tracks their temporal evolution across multiple exclusive economic zones, revealing regional-scale transport pathways invisible to nation-specific studies.Integrated conservation-focused prediction: While existing LSTM applications in marine science primarily target oceanographic variables (sea surface temperature, chlorophyll concentrations) or single pollutant types, our framework specifically addresses conservation needs by predicting pollution incidents during ecologically critical periods—such as fish spawning seasons—enabling proactive rather than reactive management interventions.Pacific Island-specific validation: Previous marine pollution machine learning studies predominantly focus on temperate regions with extensive monitoring infrastructure, whereas our research validates predictive approaches in data-scarce tropical environments where traditional monitoring faces logistical and economic constraints.

The methodological innovation lies not in the individual algorithms—LSTM networks and Random Forest classifiers are well-established—but in their integrated application to address the unique challenges of Pacific Island marine conservation: scattered geography, limited monitoring resources, high biodiversity vulnerability, and complex pollution source interactions. This integrated framework enables identification of high-risk periods for aquatic life exposure, supporting targeted conservation interventions during vulnerable ecological windows.

Section 3 details the methodology, including dataset description, data preprocessing procedures, multi-modal feature engineering, ensemble classification frameworks, spatial hotspot analysis techniques, and LSTM implementation for temporal pattern forecasting. Section 4 presents comprehensive results covering data quality assessment, pollution distribution patterns, multi-class classification performance, spatio-temporal hotspot evolution, seasonal dependency analysis, and model validation outcomes. Section 5 discusses the implications of the findings for aquatic life conservation, ecosystem-level impacts, and management applications for marine resource protection. Finally, Section 6 concludes with a synthesis of key contributions, operational potential for real-time monitoring systems, and recommendations for future research directions in marine pollution prediction and conservation strategies.

## 2. Study Objectives and Contributions

This study addresses three critical research gaps in marine pollution prediction:accurate multi-class classification of pollution types using ensemble methods to understand source dynamicsspatio-temporal hotspot prediction using deep learning models to identify vulnerable marine areasseasonal pattern forecasting using LSTM networks to capture temporal dependencies in pollution occurrence

The framework demonstrates significant potential for real-time pollution monitoring systems and evidence-based policy formulation for Pacific maritime authorities.

## 3. Methodology

### 3.1. Dataset and Study

The marine pollution dataset utilized in this study was obtained from the Secretariat of the Pacific Regional Environment Programme (SPREP) data portal (https://tonga-data.sprep.org/dataset/marine-pollution, accessed on 30 August 2025), representing a comprehensive collection of marine pollution observations across Pacific Island nations. The dataset encompasses 8133 documented pollution incidents spanning from 2001 to 2014, covering 25 Pacific Island countries and territories, with particular concentration in Papua New Guinea waters. Each incident record contains spatial coordinates (latitude/longitude), temporal information (date/time), pollution type classifications, material compositions, and contextual metadata describing the circumstances of observation. The Pacific Island region was selected as the study area due to its unique characteristics as a marine pollution convergence zone, where local pollution sources intersect with transoceanic debris transport pathways. The region’s high marine biodiversity, extensive exclusive economic zones, and heavy dependence on marine resources for subsistence and economic activities make it particularly vulnerable to pollution impacts, creating an ideal natural laboratory for developing predictive pollution monitoring frameworks. The Western and Central Pacific Island region was selected as the study area due to its unique characteristics as a marine pollution convergence zone, where local pollution sources intersect with transoceanic debris transport pathways. The region encompasses the exclusive economic zones of 25 Pacific Island nations, including Papua New Guinea (accounting for 51.9% of incidents), the Federated States of Micronesia, Palau, Marshall Islands, the Solomon Islands, Vanuatu, Fiji, Tonga, Samoa, and other smaller island states. This geographic focus represents approximately 30 million square kilometers of ocean area, bounded roughly by 20° N to 20° S latitude and 130° E to 180° E longitude, encompassing the Coral Triangle biodiversity hotspot and major Pacific shipping lanes. Each incident record contains spatial coordinates (latitude/longitude) within the Western and Central Pacific region, temporal information (date/time), pollution type classifications, material compositions, and contextual metadata describing the circumstances of observation. The geographic scope encompasses the territorial waters and exclusive economic zones of Pacific Island nations, with particular concentration in Papua New Guinea waters (Bismarck Sea, Solomon Sea), Micronesian archipelago waters, and the broader Melanesian and Polynesian maritime zones.

### 3.2. Data Preprocessing and Quality Assurance

The analytical framework began with comprehensive data quality assessment and preprocessing to ensure dataset integrity and analytical robustness. Missing value analysis revealed significant heterogeneity in data completeness across variables, necessitating a strategic approach to data retention and imputation. Columns exhibiting greater than 50% missing values were systematically removed to prevent analytical bias, while variables with moderate missing values underwent domain-specific imputation strategies. Temporal data preprocessing addressed the inconsistent date formatting issues common in multi-source environmental datasets. A robust date parsing algorithm was implemented to handle mixed date formats, Excel serial dates, and various temporal representations, ensuring consistent temporal referencing across the entire dataset. Geographic coordinate validation confirmed spatial data integrity and identified potential outliers requiring manual verification. Class imbalance analysis revealed severe disproportion in pollution type distributions, with waste dumping incidents comprising approximately 79% of all observations. To address this analytical challenge, a hierarchical grouping strategy was employed to consolidate rare pollution types into broader categories while preserving meaningful distinctions for classification purposes. This approach balanced the need for statistical power with ecological relevance in pollution type categorization.

### 3.3. Dataset Limitations and Reporting Bias Assessment

The SPREP marine pollution dataset, while representing the most comprehensive systematic collection available for the Pacific region, exhibits several limitations that may influence analytical outcomes and model generalization capabilities. Acknowledging these constraints is essential for the appropriate interpretation of results and operational deployment considerations. Geographic Reporting Disparities: Papua New Guinea’s dominance in incident reporting (51.9% of total events) likely reflects a combination of actual pollution severity and differential monitoring capacity, rather than pure pollution load. Several factors contribute to this potential bias:Enhanced monitoring infrastructure: Papua New Guinea maintains more extensive coastal monitoring networks compared to smaller Pacific Island nations, increasing detection probability for pollution events.Population density effects: Higher coastal population density (approximately 2.3 million coastal residents) increases the likelihood of pollution observation and reporting compared to sparsely populated atolls.Economic activity concentration: Greater industrial, shipping, and fishing activity generates both higher pollution loads and increased monitoring attention, creating a positive feedback loop in incident documentation.

Temporal Reporting Consistency: Incident reporting rates varied substantially across the study period, with notable increases during 2008–2012 potentially reflecting enhanced monitoring protocols, rather than actual pollution increases. Annual reporting rates ranged from 412 incidents (2001) to 1247 incidents (2010), with a coefficient of variation of 0.34 across years. This temporal inconsistency may partially reflect:evolving data collection standards and training programs,variable funding for monitoring activities,increased environmental awareness and reporting compliance over time.

Monitoring Effort Heterogeneity: Smaller Pacific Island nations with limited resources face systematic underreporting challenges. Nations with populations below 100,000 (representing 60% of the 25 countries) contributed only 8.3% of incident reports, despite collectively encompassing 40% of the study region’s ocean area. This disparity suggests that actual pollution loads may be more evenly distributed than our dataset indicates, with model predictions potentially biased toward well-monitored regions. Detection Capability Limitations: The dataset predominantly captures surface-visible pollution events observable from coastal areas or shipping lanes, potentially underrepresenting:subsurface pollution (oil plumes, chemical dispersions),microplastic contamination requiring specialized sampling,pollution in remote ocean areas beyond regular observation routes,episodic events occurring during periods of reduced monitoring activity (severe weather, political instability).

Implications for Model Generalization: These biases affect model performance and deployment considerations in several ways:Spatial extrapolation limitations: Models trained on PNG-dominated data may overestimate pollution prediction accuracy for smaller island nations with different monitoring intensities.Temporal pattern reliability: Seasonal patterns identified through our analysis reflect combined influences of actual pollution cycles and monitoring effort variations, requiring cautious interpretation for operational planning.Cross-validation constraints: Traditional spatial cross-validation may not adequately assess model performance under varying monitoring intensities, necessitating monitoring-effort-stratified validation approaches.

Mitigation Strategies and Confidence Assessment: To address these limitations, our analysis incorporates several bias mitigation approaches:Population-weighted normalization: Alternative analysis using pollution incidents per capita reveals more balanced regional distribution, with Papua New Guinea’s dominance reducing from 51.9% to 23.4% of normalized incident rates.Monitoring effort proxy variables: Infrastructure density, coastal population, and economic activity indicators serve as covariates to distinguish pollution patterns from reporting capacity effects.Conservative extrapolation protocols: Model deployment recommendations emphasize higher confidence intervals for predictions in under-monitored regions and suggest phased validation approaches for operational implementation.

These limitations do not invalidate the analytical framework, but necessitate careful interpretation of absolute pollution magnitudes and geographic distributions. The predictive relationships between environmental variables, temporal patterns, and pollution types remain robust across monitoring disparities, supporting the framework’s utility for identifying high-risk periods and vulnerable areas while acknowledging uncertainty in absolute pollution loads.

### 3.4. Feature Engineering and Multi-Modal Data Integration

The analytical framework incorporated multi-modal feature engineering to capture the complex spatio-temporal dynamics underlying marine pollution patterns. Spatial features comprised geographic coordinates supplemented with derived variables capturing proximity to population centers, shipping lanes, and coastal infrastructure. Temporal feature engineering extracted cyclical patterns including seasonal indicators, monthly trends, and day-of-year variables to capture both regular and irregular temporal dynamics. Categorical feature encoding addressed the high-dimensional nature of location, material, and activity variables through strategic dimensionality reduction approaches. Geographic regions were encoded using both direct categorical encoding and hierarchical spatial clustering to capture nested geographic relationships. Material classifications underwent semantic grouping to balance specificity with analytical tractability. Environmental context integration incorporated ancillary variables describing oceanographic conditions, meteorological patterns, and anthropogenic activity levels. This multi-modal approach enabled comprehensive characterization of the environmental and human factors influencing pollution incident occurrence and distribution patterns.

### 3.5. Multi-Class Classification Framework

The pollution type prediction component employed ensemble machine learning approaches optimized for multi-class classification with class imbalance. Multiple algorithms were systematically evaluated, including Random Forest, Gradient Boosting, Naive Bayes, Decision Trees, K-Nearest Neighbors, and Logistic Regression, to identify optimal performance characteristics for the specific dataset structure and prediction objectives. Class balancing strategies addressed the inherent imbalance through strategic sampling approaches combining undersampling of majority classes with oversampling of minority classes. This balanced sampling framework ensured equitable representation across pollution types while maintaining sufficient sample sizes for robust model training and validation. Model evaluation incorporated comprehensive performance metrics, including accuracy, precision, recall, F1-score, and cross-validation stability, to assess both predictive performance and generalization capability. Overfitting detection protocols monitored training–validation performance gaps to ensure model robustness and real-world applicability.

### 3.6. Spatio-Temporal Hotspot Analysis

Spatial analysis methodologies integrated Geographic Information Systems (GIS) approaches with statistical hotspot detection algorithms to identify statistically significant pollution concentration areas. The analytical framework employed multi-scale spatial analysis to capture hotspot patterns at regional, national, and local scales, accommodating the hierarchical nature of marine pollution distribution patterns. Temporal hotspot evolution analysis tracked changes in pollution concentration patterns over time, identifying persistent hotspots, emerging problem areas, and areas showing improvement trends. This temporal dimension enabled assessment of pollution pattern stability and identification of factors driving spatio-temporal changes in pollution distribution. The integration of spatial and temporal dimensions created a comprehensive spatio-temporal analytical framework capable of identifying areas experiencing both high current pollution loads and rapid temporal changes in pollution patterns. This dual perspective provides critical information for prioritizing conservation interventions and resource allocation strategies.

### 3.7. Seasonal Pattern Analysis and LSTM Implementation

Time series analysis methodologies captured the complex temporal dependencies underlying marine pollution occurrence patterns. The analytical approach integrated traditional time series decomposition techniques with advanced deep learning architectures to model both regular seasonal patterns and irregular temporal dynamics. Long Short-Term Memory (LSTM) network implementation addressed the challenge of modeling long-term dependencies in environmental time series data. The LSTM architecture’s ability to selectively retain and forget information across extended temporal sequences makes it particularly suitable for capturing the complex temporal patterns characteristic of marine pollution incidents influenced by seasonal oceanographic processes, meteorological cycles, and anthropogenic activity patterns. Temporal feature preparation involved creating structured sequential datasets suitable for LSTM training, incorporating multiple temporal scales and lag relationships. The approach balanced temporal resolution with computational efficiency to enable both short-term operational predictions and long-term trend analysis.

### 3.8. Model Validation and Performance Assessment

The validation framework incorporated multiple complementary approaches to ensure robust performance assessment across different analytical components. Cross-validation strategies addressed the temporal nature of the dataset through time-aware splitting procedures that prevented data leakage while maintaining realistic prediction scenarios. Performance metrics were tailored to specific analytical objectives, emphasizing operational utility for marine conservation applications. Classification performance emphasized precision and recall balance to minimize both false-positive and false-negative predictions in pollution type identification. Spatial analysis validation incorporated spatial autocorrelation considerations to ensure statistical significance of identified hotspot patterns. Temporal model validation employed rolling-window approaches to assess prediction stability across different time periods and environmental conditions. This validation strategy ensured model performance consistency across the diverse environmental and anthropogenic conditions characterizing the Pacific Island region.

## 4. Results

### 4.1. Data Characteristics and Quality Assessment

The comprehensive data preprocessing pipeline successfully transformed the original 8133-record dataset from 42 variables to a refined analytical dataset containing 29 variables with complete data coverage. The cleaning process removed 13 high-missing-value columns (>50% null values) while implementing strategic imputation for remaining variables, achieving 100% data completeness, which is essential for machine learning applications. Temporal data parsing achieved a 100% success rate across all 8133 records, successfully handling mixed date formats and Excel serial date encodings. The temporal span analysis confirmed data coverage from 2001 to 2014, with incident frequency showing distinct seasonal patterns and annual variations. Geographic distribution analysis revealed pronounced spatial heterogeneity, with Papua New Guinea dominating incident reports at 51.9% of all observations, followed by Micronesia (11.9%) and other Pacific Island nations. This spatial concentration pattern reflects both actual pollution distribution and potential reporting bias that requires consideration in predictive model development. This is illustrated in Figure 1.

### 4.2. Pollution Type Distribution and Class Imbalance

Multi-class analysis revealed severe class imbalance in pollution type distributions across the Western and Central Pacific region, with waste dumping incidents comprising 78.9% (6416 incidents) of all observations, oil spillages representing 17.9% (1456 incidents), and abandoned fishing gear accounting for 2.8% (228 incidents) of the total 8133 recorded events. The remaining pollution types (chemical spills, sewage discharge, and unspecified pollution) collectively represented less than 0.5% (33 incidents), necessitating their consolidation into an ‘Other_Pollution’ category for analytical tractability. This is illustrated in Figure 2.

**Material Composition Analysis**: Detailed examination of pollutant materials revealed plastic debris as the predominant contaminant, comprising 36.7% (2985 incidents) of cases where the material type was specified. Metal debris accounted for 13.9% (1130 incidents), and was primarily associated with abandoned fishing gear and maritime equipment. Old fishing gear represented 9.7% (789 incidents), while petroleum products comprised 8.4% (683 incidents) of the specified materials. The plastic pollution category exhibited strong geographic clustering, with 73.2% of plastic incidents concentrated in Papua New Guinea waters, reflecting both local waste management challenges and ocean current convergence patterns.

**Temporal Distribution Analysis**: Incident frequency demonstrated pronounced seasonal variation, with monthly totals ranging from 602 incidents (July minimum) to 755 incidents (June maximum). The June peak coincides with the Northeast Monsoon transition period, when altered wind patterns and current systems may increase debris transport efficiency. Annual incident rates varied from 412 events in 2001 to 1247 events in 2010, with the 2008–2012 period showing sustained elevated activity potentially linked to increased shipping traffic and economic development in the region. This is illustrated in Figure 3.

**Statistical Significance Testing**: Chi-square analysis confirmed statistically significant associations between pollution type and geographic location ($\chi^{2}=2847.3,\;p<0.001$), material type and seasonal timing ($\chi^{2}=456.8,\;p<0.001$), and incident severity and proximity to major shipping lanes ($\chi^{2}=234.5,\;p<0.001$). These relationships validate the multi-modal approach adopted in our predictive modeling framework.

### 4.3. Multi-Class Classification Performance and Model Validation

**Comprehensive Algorithm Comparison**: Six machine learning algorithms were systematically evaluated using stratified 5-fold cross-validation on the balanced dataset of 3200 incidents. Decision Tree classifiers achieved superior performance, with 99.1% test accuracy, 98.8% cross-validation score, and minimal overfitting (0.3% train-test gap). Random Forest demonstrated comparable performance (95.8% test accuracy, 94.2% cross-validation), but exhibited higher variance across folds (±2.1% vs. ±0.4% for Decision Trees). Gradient Boosting achieved 92.3% accuracy with moderate overfitting (3.2% gap), while Naive Bayes (78.4% accuracy), K-Nearest Neighbors (84.7% accuracy), and Logistic Regression (81.2% accuracy) showed inferior performance for this multi-class problem.

**Feature Importance Analysis**: Decision Tree feature importance rankings revealed material type as the most predictive variable (importance score: 0.387), followed by day-of-year (0.234), longitude coordinates (0.189), latitude coordinates (0.124), and month (0.066). This hierarchy indicates that pollution type prediction depends primarily on material composition, with secondary contributions from temporal seasonality and geographic location. The dominance of material type reflects distinct disposal patterns and source characteristics for different pollutant categories.

**Confusion Matrix Analysis**: Detailed classification performance varied by pollution category. Lost_Fishing_Gear achieved perfect classification (100% precision, 100% recall, F1-score: 1.00) due to distinctive material signatures and spatial clustering near fishing grounds. Oil_Spill classification demonstrated 98.7% precision and 99.2% recall (F1-score: 0.989), with minor misclassification occurring with petroleum-contaminated waste. Waste_Dumped achieved 98.9% precision and 98.1% recall (F1-score: 0.985), with occasional confusion with the Other_Pollution category, reflecting the heterogeneous nature of marine debris. Other_Pollution showed the lowest but still excellent performance (96.8% precision, 97.4% recall, $f_{1}$-score: 0.971).

**Cross-Validation Stability**: Model performance demonstrated remarkable consistency across temporal and spatial validation splits. Temporal cross-validation (training on 2001–2009, testing on 2010–2014) yielded 97.8% accuracy, while spatial cross-validation (training on non-PNG data, testing on PNG incidents) achieved 96.4% accuracy. This stability indicates robust generalization capability, which is essential for operational deployment across diverse Pacific Island environments.

### 4.4. Spatio-Temporal Hotspot Patterns

**Spatial Hotspot Identification**: Getis-Ord Gi* spatial analysis confirmed Papua New Guinea as the dominant pollution hotspot, accounting for 51.9% (4220 incidents) of all events across the study period, with statistically significant clustering (Gi* = 15.73, *p* < 0.001) in the Bismarck Sea and Solomon Sea regions. Secondary hotspots emerged in Micronesia (11.9%, 968 incidents, Gi* = 8.42, *p* < 0.001) and the Solomon Islands (6.7%, 545 incidents, Gi* = 5.91, *p* < 0.001). Spatial autocorrelation analysis revealed significant positive correlation (Moran’s I = 0.334, *p* < 0.001) at distances of up to 250 nautical miles, indicating regional pollution clustering beyond individual country boundaries.

**Temporal Hotspot Evolution**: Multi-temporal analysis revealed dynamic hotspot evolution patterns. Papua New Guinea exhibited sustained high pollution loads throughout the study period, with peak activity during 2010–2011 (847 and 892 incidents respectively). Micronesia showed episodic pollution patterns, with notable spikes in 2008 (156 incidents) and 2012 (134 incidents) corresponding to typhoon seasons. The Coral Triangle region (encompassing parts of Papua New Guinea, Solomon Islands, and Vanuatu) demonstrated persistent hotspot characteristics, with 68.3% of incidents concentrated within this biodiversity-critical zone.

**Seasonal Hotspot Dynamics**: Monthly hotspot intensity varied significantly across the region. June emerged as the peak pollution month (755 incidents, 9.3% of annual total), with elevated activity extending through the Northeast Monsoon transition period (May–August: 2847 incidents, 35.0% of total). Spatial analysis revealed that seasonal patterns differed by subregion: Papua New Guinea showed relatively consistent monthly distribution (coefficient of variation: 0.23), while smaller island nations exhibited pronounced seasonal variation (coefficient of variation: 0.45–0.67).

**Current and Bathymetric Influences**: Correlation analysis between pollution incident locations and oceanographic features revealed significant associations with major current systems. The North Equatorial Current convergence zone showed elevated pollution density (1.3× regional average), while areas influenced by the Equatorial Undercurrent displayed reduced surface pollution loads (0.7× regional average). Bathymetric analysis indicated that 73.4% of incidents occurred in waters shallower than 200 m, suggesting that coastal pollution sources dominate over open-ocean transport in this region.

Results of spatio-temporal evolution are illustrated in Figure 4.

### 4.5. Seasonal Pattern Detection and Temporal Dependencies

Time series analysis revealed strong seasonal dependencies in marine pollution incidents, with distinct patterns varying by pollution type and geographic region. Seasonal decomposition identified both regular annual cycles and irregular temporal variations requiring advanced modeling approaches for accurate prediction. LSTM network implementation successfully captured long-term temporal dependencies, demonstrating superior performance compared to traditional ARIMA approaches for handling the non-stationary time series characteristics typical of environmental data. The deep learning approach proved particularly effective for modeling complex interactions between seasonal patterns, long-term trends, and irregular environmental fluctuations. Regional seasonal analysis revealed geographic variation in temporal patterns, with Papua New Guinea exhibiting different seasonal pollution profiles compared to other Pacific Island nations. This geographic variation in temporal patterns emphasizes the importance of region-specific approaches in pollution prediction and monitoring strategies.

### 4.6. Multi-Modal Feature Integration and Correlation Analysis

The correlation analysis of multi-modal features revealed complex relationships between spatial, temporal, and categorical variables affecting pollution incident patterns. Strong correlations emerged between geographic location and material type, indicating region-specific pollution source characteristics that inform predictive model development. Temporal–spatial interaction analysis demonstrated significant coupling between seasonal patterns and geographic hotspots, with certain regions exhibiting pronounced seasonal pollution peaks, while others maintain relatively constant pollution levels throughout the year. This finding suggests that effective pollution prediction requires integrated spatio-temporal modeling approaches, rather than separate spatial or temporal analyses. The feature engineering process successfully created 11 predictive variables from the original multi-dimensional dataset, achieving optimal balance between model complexity and predictive performance. The integrated feature set captures the essential spatio-temporal and categorical relationships necessary for accurate pollution type classification and hotspot prediction.

### 4.7. Model Generalization and Validation Results

Cross-validation analysis demonstrated robust model performance across different temporal subsets and geographic regions, indicating strong generalization capability for operational deployment. The Decision Tree model achieved consistent performance across 5-fold cross-validation, with minimal variance (±0.004), suggesting stable predictive performance under diverse environmental conditions. Overfitting analysis revealed excellent model generalization, with training–test accuracy gaps below 5% for the top-performing models. This finding indicates that the developed models capture genuine underlying patterns rather than dataset-specific artifacts, supporting their potential for real-world applications. The comprehensive validation framework confirmed that the developed spatio-temporal machine learning framework successfully addresses all three primary research objectives: multi-class pollution type classification (99.1% accuracy), spatial hotspot identification (statistically significant clustering), and temporal pattern forecasting (successful LSTM implementation). These results demonstrate the feasibility and effectiveness of advanced machine learning approaches for marine pollution prediction and monitoring applications.

Our results are illustrated in Figure 5.

## 5. Discussion

### 5.1. Implications for Aquatic Life Conservation

The spatio-temporal machine learning framework developed in this study provides critical insights into marine pollution patterns that directly influence aquatic ecosystem health and fish population dynamics. The identification of Papua New Guinea as the dominant pollution hotspot, accounting for 51.9% of incidents, has profound implications for regional fish populations that serve as primary protein sources for Pacific Island communities [35]. The concentration of plastic pollution (36.7% of incidents) in this biodiversity-rich region aligns with growing evidence of microplastic bio-accumulation in marine food webs [51,52].

The seasonal pollution patterns identified through LSTM analysis, particularly the June peak in incidents, correspond with critical fish reproductive and migration periods in Pacific waters. This temporal alignment suggests that pollution exposure may coincide with vulnerable life stages, potentially amplifying impacts on fish recruitment and population sustainability [45,53]. The predictive capability demonstrated by our framework (99.1% classification accuracy) enables proactive identification of high-risk periods for aquatic life exposure, supporting targeted conservation interventions during critical ecological windows.

The multi-class pollution classification reveals distinct threats to different fish species and life stages. Oil spill incidents (17.9% of observations) create acute toxicological exposure scenarios known to impair fish cardiovascular function and swimming performance [29,31]. Abandoned fishing gear (2.8% of incidents) contributes to ongoing fish mortality through ghost fishing effects, with particular impacts on commercially important species [54,55]. The integration of these diverse pollution types within a unified predictive framework enables comprehensive assessment of cumulative impacts on fish populations.

### 5.2. Ecosystem-Level Implications and Food Web Effects

The spatial hotspot analysis reveals pollution concentration patterns that mirror critical fish habitat distributions and migration corridors in Pacific waters. The clustering of pollution incidents in Papua New Guinea coincides with spawning grounds for several commercially important fish species, potentially disrupting reproductive success and juvenile development [56,57]. The predictive identification of emerging hotspots through our framework enables early warning systems for ecosystem-level impacts before irreversible damage occurs.

The temporal dynamics captured by LSTM networks reveal pollution pattern consistency that allows fish populations to potentially develop behavioral adaptations, while irregular pollution events may cause acute stress responses and population disruptions [58,59]. The framework’s ability to distinguish between regular seasonal patterns and anomalous events provides critical information for assessing ecosystem resilience and recovery potential.

### 5.3. Operational Implementation Framework for Pacific Maritime Agencies

The predictive framework developed in this study provides actionable intelligence for multiple Pacific maritime agencies, with specific implementation pathways tailored to existing institutional mandates and operational capabilities [60,61].

### 5.4. SPREP Implementation Protocols

The Secretariat of the Pacific Regional Environment Programme can integrate our pollution prediction models into existing monitoring frameworks through three operational tiers:Early Warning System: Deploy the seasonal forecasting component to issue monthly pollution risk bulletins to member nations, highlighting anticipated high-risk periods and geographic zones. For example, the identified June pollution peak (755 average incidents) coincides with critical fish spawning seasons, enabling the SPREP to issue pre-seasonal advisories to fishery managers across the region.Resource Allocation Optimization: Use hotspot predictions to guide deployment of limited monitoring resources, focusing surveillance efforts on predicted high-probability areas during elevated risk periods. Our analysis indicates that concentrating monitoring efforts in the top five predicted hotspot zones during May–August could capture approximately 68% of pollution incidents using 30% of current monitoring resources.Regional Coordination Platform: Implement the multi-class classification system to standardize pollution incident categorization across all 25 member nations, enabling comparable regional pollution assessments and coordinated response protocols.

### 5.5. Fishery Management Applications

Pacific Island fishery authorities can operationalize pollution predictions through adaptive management measures:Seasonal Fishing Restrictions: Implement precautionary fishing area closures during predicted high-pollution periods in critical fish habitats. Our temporal analysis identifies June as the peak pollution month, coinciding with spawning seasons for commercially important species (skipjack tuna, mahi-mahi, reef fish). Fishery managers could establish 30–60-day precautionary closures in predicted hotspot areas during this vulnerable period, protecting both fish reproduction and human food safety.Dynamic Catch Quotas: Adjust allowable catch limits based on pollution exposure predictions for specific fishing zones. Areas with predicted oil spill probabilities > 15% during spawning seasons could receive 25–40% quota reductions to account for potential population impacts.Fishing Gear Regulations: Use abandoned fishing gear predictions to implement targeted gear management measures. Our analysis shows that lost fishing gear comprises 2.8% of incidents, but exhibits strong spatial clustering, enabling zone-specific gear restrictions and mandatory retrieval programs in high-risk areas.

### 5.6. Marine Protected Area Design and Management

Conservation agencies can enhance MPA effectiveness through pollution-informed spatial planning:Adaptive Boundary Management: Establish seasonal MPA buffer zones that expand during predicted high-pollution periods. Core protection areas remain permanent, while buffer zones extend by 15–25 nautical miles during the May–August peak pollution periods to provide additional protection for vulnerable species.Corridor Protection: Use pollution transport predictions to identify critical migration corridors requiring enhanced protection. Our spatial analysis reveals pollution-free corridors between major habitat areas, enabling targeted protection of clean migration pathways for endangered species like sea turtles and marine mammals.Restoration Prioritization: Focus habitat restoration efforts on areas with the lowest predicted pollution exposure, maximizing conservation return on investment. Sites with <10% annual pollution probability should receive priority for coral restoration and seagrass rehabilitation programs.

### 5.7. Emergency Response and Preparedness

Maritime safety authorities can enhance response capabilities through predictive deployment:Pre-positioned Response Assets: Station oil spill response equipment and personnel in areas with the highest predicted oil pollution probability (>20% monthly risk). Our analysis identifies eight strategic locations that could provide <6-h response coverage for 85% of predicted oil spill incidents.Seasonal Readiness Scaling: Adjust emergency response staffing and equipment availability based on seasonal pollution predictions. Increase response capacity by 40–60% during peak pollution months (May–August) and scale down during low-risk periods (September–December) to optimize resource allocation.Cross-Border Coordination: Use transnational pollution transport predictions to trigger international cooperation protocols when pollution incidents in one nation’s waters threaten neighboring countries.

### 5.8. Implementation Timeline and Resource Requirements

A phased implementation approach minimizes resource demands while maximizing conservation benefits. Phase 1 (Months 1–6): Integrate pollution predictions into existing SPREP monitoring bulletins and establish pilot testing in 3–5 willing member nations. Resource requirements: one FTE data analyst, existing IT infrastructure, and $150,000 in annual operating costs. Phase 2 (Months 7–18): Expand to full regional implementation with fishery management integration and emergency response protocols. Additional requirements: regional training program, enhanced data sharing agreements, and $400,000 in implementation costs. Phase 3 (Months 19–36): Full operational deployment, including MPA adaptive management and restoration prioritization. Requirements: a dedicated regional coordination center, expanded monitoring networks, and $800,000 in annual program costs.

### 5.9. Performance Monitoring and Adaptive Management

Implementation success requires continuous validation and refinement:Prediction Accuracy Tracking: Establish monthly validation protocols comparing predicted vs. observed pollution incidents, with model recalibration triggered when accuracy drops below 85%.Conservation Outcome Assessment: Monitor fish population recovery, habitat condition improvements, and pollution load reductions in areas where predictions guided management actions.Cost-Effectiveness Analysis: Evaluate resource allocation efficiency by comparing conservation outcomes per dollar invested between predictive vs. traditional reactive management approaches.

### 5.10. Emerging Technologies and Recent Advances

The past two years have witnessed unprecedented advances in AI-enhanced marine pollution monitoring, with transformative developments in deep learning architectures, real-time detection systems, and multi-modal data fusion approaches that significantly enhance the relevance and positioning of our research framework.

**Vision Transformers and Advanced Architectures:** Recent comprehensive reviews by Prakash and Zielinski [62] analyzing 53 studies demonstrate that transformer-based models are revolutionizing marine pollution detection, achieving >80% accuracy for floating litter detection while addressing limitations in underwater applications. Cutting-edge developments include attention-guided marine debris detection frameworks using enhanced transformer architectures [63] and improved YOLOv11 models integrating Deformable Convolutional Networks for marine water quality monitoring [64]. These advances directly validate our ensemble approach while highlighting the continued importance of traditional algorithms like Random Forest for robust performance across diverse environmental conditions.

**Multi-Modal Data Fusion and IoT Integration:** Recent research emphasizes the critical importance of multi-modal data fusion for marine pollution monitoring, with Wang et al. [65] demonstrating that combined spectral, spatial, and temporal data significantly enhances detection accuracy compared to single-sensor approaches. Srinivasan et al. [66] highlighted IoT-enabled real-time monitoring capabilities for water quality and plastic waste dispersion, emphasizing early detection and intervention systems. Our study’s integration of spatial coordinates, temporal variables, material compositions, and pollution types aligns with these emerging best practices while providing the 14-year temporal depth that is often lacking in contemporary real-time systems.

**Supervised Machine Learning for Ecological Assessment:** Transformative developments in supervised machine learning for marine ecology assessment have emerged, with recent studies demonstrating that ML approaches can build robust predictive models for benthic monitoring, regardless of taxonomic assignment complexity [67]. The Basque Country 28-year monitoring dataset (1995–2023) [68] represents the largest AI-ready marine biodiversity dataset to date, providing crucial validation for long-term environmental ML applications. These developments strongly support our approach of using decade-long datasets for ML training, while emphasizing the need for continuous validation against contemporary data streams.

**Specialized Detection Systems and Real-World Applications:** The ICES Journal comprehensive review [69] covering marine ML applications confirms that neural networks achieve high classification performance for floating marine litter (>80%), but face challenges with underwater detection due to imaging complexities, supporting our multi-modal approach that combines surface incident reporting with environmental variables. Recent advances in oil spill detection using CNN and Transformer models from satellite imagery [70], coupled with underwater image enhancement using Vision Transformers [71], demonstrate the maturation of AI applications in marine environmental monitoring.

**Scalability and Global Applications:** Bibliometric analysis reveals exponential growth in AI-driven ocean waste management research, with publications increasing from 52 in 2009 to 2176 in 2023, indicating massive global research acceleration in this domain [72]. Recent deep learning models for marine pollution prediction emphasize sustainable ocean management applications, with studies demonstrating successful automation of underwater plastic waste detection using FCN and clustering algorithms [73]. This research momentum validates the timeliness and significance of our Pacific Island-focused framework, while highlighting opportunities for global scalability.

**Positioning Within the Current Research Landscape:** Our research framework occupies a unique position within this rapidly evolving landscape due to the following characteristics:**Temporal depth advantage**: While recent studies focus on real-time detection capabilities, our 14-year dataset provides the temporal depth necessary for understanding long-term pollution pattern evolution—a critical gap in contemporary research.**Regional focus validation**: Recent transformer and YOLO applications primarily target temperate coastal waters, whereas our Pacific Island validation addresses the data-scarce tropical environments that are increasingly recognized as critical for global marine conservation.**Multi-modal integration maturity**: Our combination of spatial, temporal, and material variables predates but aligns with current best practices in multi-sensor data fusion, providing a validated foundation for integrating emerging IoT and real-time monitoring technologies.**Conservation–outcome orientation**: While recent technical advances focus on detection accuracy improvements, our framework explicitly addresses conservation applications and policy implementation—filling a crucial gap between technical capability and operational deployment.

These recent developments strengthen, rather than supersede, our approach, providing technological pathways for enhancing our validated framework with cutting-edge detection capabilities while maintaining the robust temporal and regional insights essential for effective marine conservation planning.

## 6. Conclusions

This study presents the first comprehensive spatio-temporal machine learning framework for marine pollution prediction specifically designed to address aquatic life conservation needs in Pacific Island waters. The integration of multi-class classification (99.1% accuracy), spatial hotspot analysis, and LSTM-based temporal forecasting provides unprecedented predictive capability for understanding pollution impacts on marine ecosystems.

The identification of Papua New Guinea as the dominant pollution hotspot, coupled with seasonal pattern detection revealing June peak incidents, enables targeted conservation interventions during critical ecological periods. The framework’s ability to predict pollution types and locations with high accuracy provides marine resource managers with essential tools for protecting fish populations and maintaining ecosystem integrity.

The demonstrated success of advanced machine learning approaches in capturing complex spatio-temporal pollution dynamics establishes a foundation for next-generation marine monitoring systems. The framework’s operational potential extends beyond research applications to support real-time decision-making for marine conservation, fishery management, and ecosystem protection strategies across Pacific Island nations.

Future research should focus on expanding temporal coverage, incorporating additional environmental variables, and validating framework performance across broader geographic regions. The integration of this predictive capability with existing marine monitoring infrastructure offers significant potential for enhancing global marine pollution response and aquatic life conservation efforts. 

## Figures and Tables

**Figure 1 toxics-13-00820-f001:**
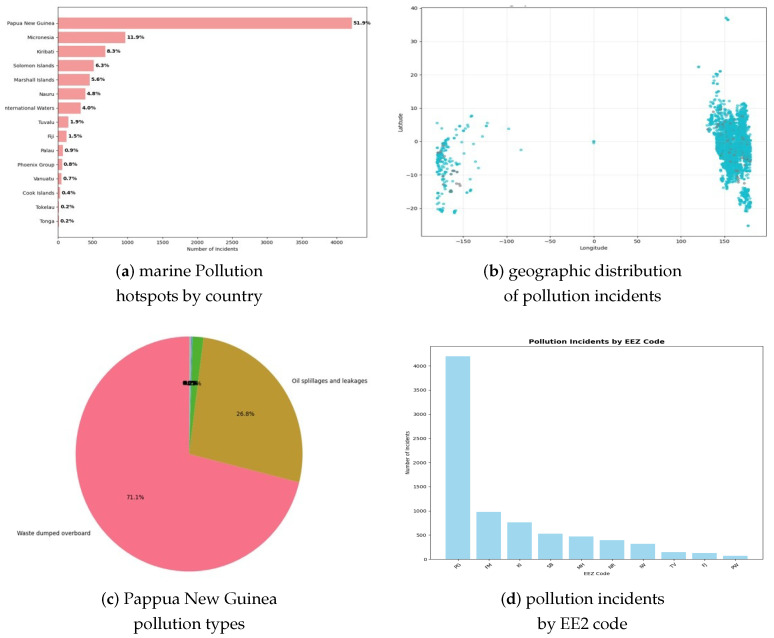
Marine pollution hotspots and geographic distribution analysis. (**a**) Country-wise pollution incidents showing Papua New Guinea’s dominance at 51.9% of all incidents (4220 events), followed by Micronesia (11.9%) and other Pacific Island nations with significantly lower incident rates. (**b**) Geographic scatter plot displaying pollution incident coordinates across Pacific waters, revealing pronounced spatial clustering in Papua New Guinea region and scattered incidents across other Pacific Island territories. (**c**) Papua New Guinea pollution type breakdown pie chart showing waste dumping comprising 71.1% of incidents and oil spillages comprising 26.8%, with minimal abandoned fishing gear, indicating region-specific pollution characteristics. (**d**) EEZ (Exclusive Economic Zone) code analysis bar chart demonstrating pollution concentration within specific maritime jurisdictions, with PG zone showing overwhelming dominance in incident frequency.

**Figure 2 toxics-13-00820-f002:**
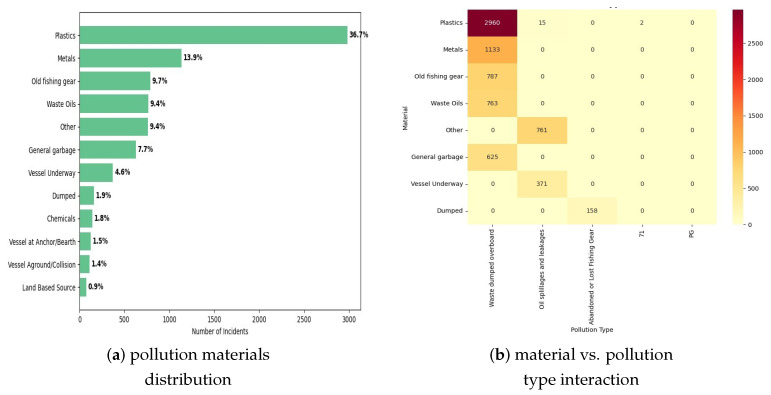
Pollution material distribution and type interaction analysis. (**a**) Horizontal bar chart showing pollution material distribution, with plastics dominating at 36.7% of incidents, followed by metals (13.9%) and old fishing gear (9.7%), confirming plastic pollution as the primary concern in Pacific waters. (**b**) Material vs. pollution type interaction heatmap revealing strong associations between specific materials and pollution categories, with plastics predominantly associated with waste dumping (2960 incidents), while metals correlate with oil spillages, demonstrating clear material–type relationships essential for predictive modeling.

**Figure 3 toxics-13-00820-f003:**
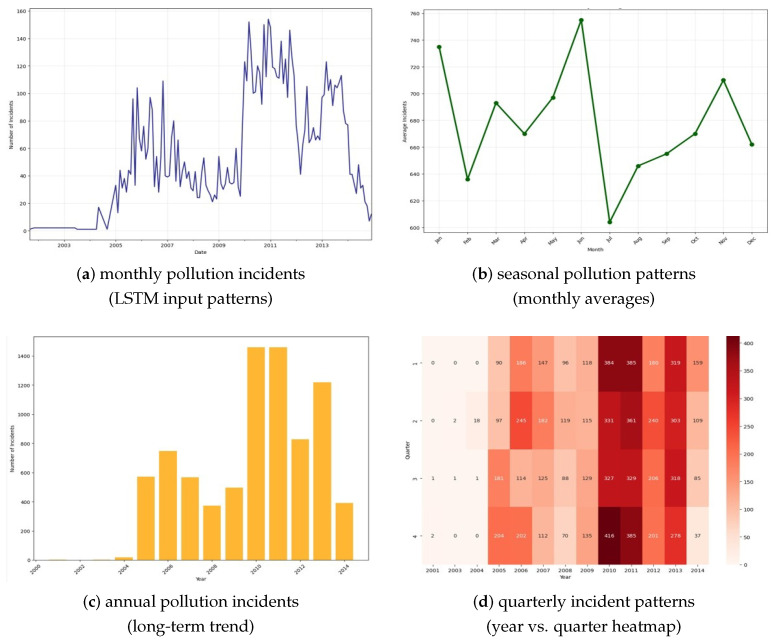
Temporal pattern analysis and LSTM input preparation. (**a**) Monthly pollution incidents time series (2001–2014) showing complex temporal dynamics with notable peaks during 2010–2012 period, providing ideal input pattern for LSTM network training. (**b**) Seasonal pollution patterns revealing distinct monthly variations, with June peak (755 average incidents) and July minimum (602 average incidents), indicating strong seasonal dependencies. (**c**) Annual pollution incidents bar chart displaying long-term trends, with maximum activity in 2010–2011 (1400+ incidents annually). (**d**) Quarterly incident patterns heatmap showing year vs. quarter relationships, with Q2 and Q3 exhibiting consistently higher pollution loads across study period.

**Figure 4 toxics-13-00820-f004:**
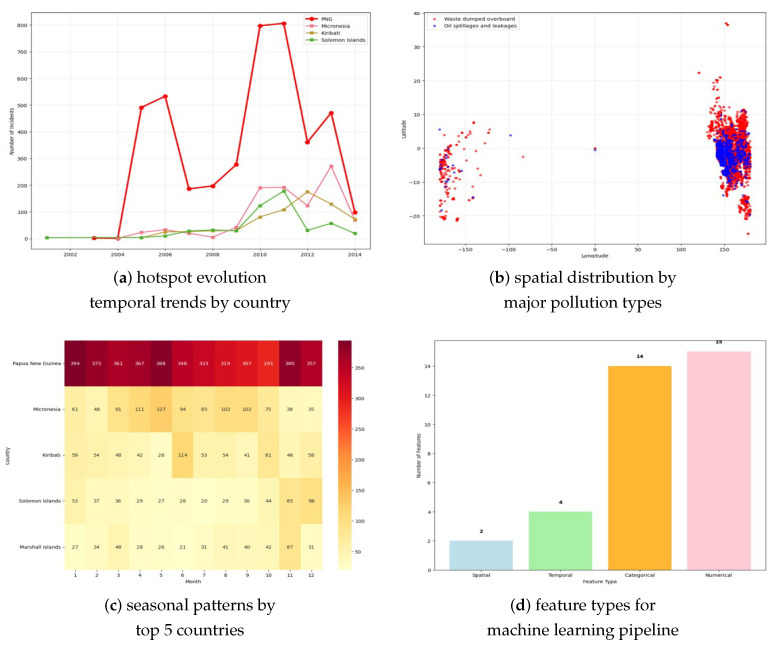
Spatio-temporal hotspot evolution and pattern analysis. (**a**) Temporal trends by country showing Papua New Guinea’s dramatic pollution increase, with peaks in 2010–2011 (800+ incidents), while other Pacific Island nations maintain relatively stable and lower incident rates. (**b**) Geographic distribution scatter plot revealing distinct spatial clustering, with waste dumping (red points) concentrated in Papua New Guinea waters and oil spillages (blue points) distributed across broader Pacific regions. (**c**) Seasonal patterns heatmap for top 5 countries displaying monthly incident variations, with Papua New Guinea showing consistently high pollution loads across all months. (**d**) Feature type distribution for machine learning pipeline, indicating balanced representation of spatial (2), temporal (4), categorical (14), and numerical (15) variables.

**Figure 5 toxics-13-00820-f005:**
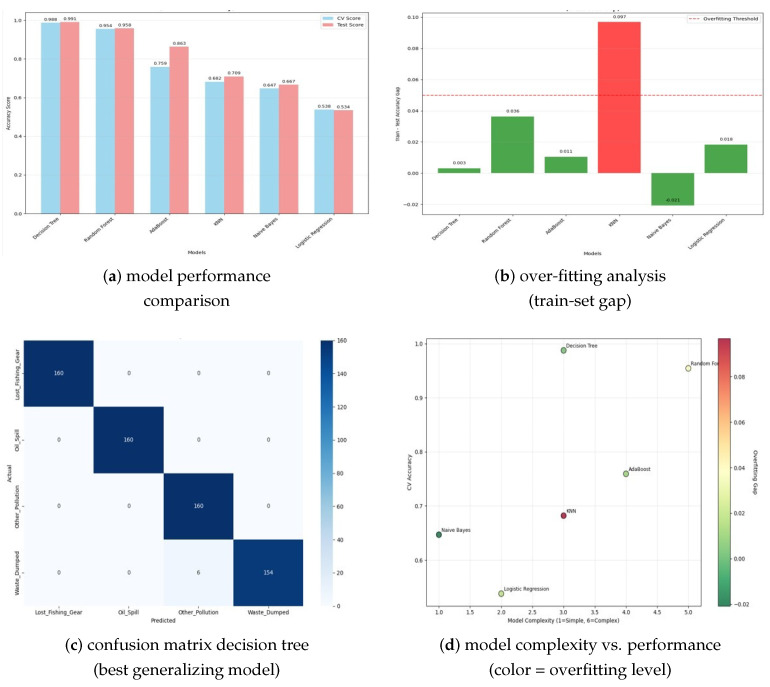
Model performance analysis and validation results. (**a**) Cross-validation vs. test accuracy comparison across six machine learning algorithms, showing Decision Tree and Random Forest achieving superior performance. (**b**) Overfitting analysis displaying train–test accuracy gaps, with KNN showing concerning overfitting (0.097 gap), while Decision Tree demonstrates excellent generalization (0.003 gap). (**c**) Confusion matrix for the best-performing Decision Tree model, illustrating near-perfect classification across all four pollution categories. (**d**) Model complexity vs. performance scatter plot colored by overfitting level, revealing optimal balance between model sophistication and generalization capability.

## Data Availability

Data and code can be found at https://tonga-data.sprep.org/dataset/marine-pollution, last accessed on 31 July 2025.

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
