# Peer review of "Spatio-Temporal Machine Learning for Marine Pollution Prediction: A Multi-Modal Approach for Hotspot Detection and Seasonal Pattern Analysis in Pacific Waters"

_toxics, 2025, doi:10.3390/toxics13100820_

Round 1

Reviewer 1 Report

Comments and Suggestions for Authors

The manuscript "Spatio-Temporal Machine Learning for Marine Pollution Prediction: A Multi-Modal Approach for Hotspot Detection and Seasonal Pattern Analysis in PacificWaters" is very large and difficult to read. The authors write in general phrases, but I lack specifics. Also, after reading, I still do not understand what type of manuscript this is? A research article or a review?

The manuscript may be published in a journal after major revision and re-review.

1. At the moment it is not clear what kind of pollution we are talking about. Plastic? Chemical or biological pollution? Authors should clearly define the subject and object of the study. This is currently quite unclear.
2. Section 2 shows common facts. I recommend that the authors remove this section and in the introduction lead the reader specifically to spatiotemporal machine learning for environmental assessment, and then the purpose of the work appears.
3. Why is the data limited to 2014? What does information from 10 years ago give?
4. The authors talk about the Pacific Ocean, but the work is about a part of the Asia-Pacific region. Here, too, it is necessary to be more specific.
5. The results are described very superficially and require more depth and detail.

Author Response

We would like to thank you for your advice and suggestions. Our responses and summary of changes are in the enclosed file

Reviewer 2 Report

Comments and Suggestions for Authors
  1. This manuscript presents a well-structured, timely, and technically strong paper that integrates machine learning, spatial analysis, and temporal modeling to address the prediction of marine pollution. However, before the paper is considered for acceptance, some revisions are required.
  2. Regarding the abstract, it is slightly overloaded with methods. It is suggested that the author could emphasize key results and contributions, not only techniques.
  3. The introduction highlights the importance of spatio-temporal prediction but does not clearly differentiate this study from previous uses of LSTM and random forests in environmental monitoring. It is recommended that the author explicitly state the innovation—whether it is the dataset size, multimodal feature integration, or the focus on the Pacific region.
  4. The dataset covers 2001–2014 (SPREP). There is no mention of potential reporting bias, such as Papua New Guinea's dominance possibly reflecting monitoring effort rather than just pollution severity. It is recommended that the author include a section acknowledging dataset limitations, reporting variability, and possible effects on model generalization.
  5. While the paper claims substantial implications for marine conservation, the translation of predictions into policy action is underdeveloped. It is suggested that the author could expand Section 5 to include concrete examples of how agencies (e.g., SPREP, fisheries regulators) could apply the model outputs in practice (e.g., seasonal bans, MPA design, gear management).
  6. The review is comprehensive. It is suggested that the author could integrate recent research on machine learning in marine ecology and pollution monitoring. This would strengthen positioning in the current research landscape.
  7. Figures are informative but dense. For example, Figures 2 and 4 could benefit from clearer legends and larger fonts. It is suggested that the author could enlarge the labels, simplify overlapping scatter plots, and ensure that color codes are explained in the captions.

Author Response

Thank you for your advice and suggestions. We have revised our manuscript accordingly. The changes are summarized in the enclosed file

Round 2

Reviewer 1 Report

Comments and Suggestions for Authors

The authors have improved the manuscript based on the reviewers' comments. I recommend acceptance of the manuscript for publication.

Reviewer 2 Report

Comments and Suggestions for Authors

No further comments to the revision.